# Performance of a Mobile 3D Camera to Evaluate Simulated Pathological Gait in Practical Scenarios

**DOI:** 10.3390/s23156944

**Published:** 2023-08-04

**Authors:** Diego Guffanti, Daniel Lemus, Heike Vallery, Alberto Brunete, Miguel Hernando, Herwin Horemans

**Affiliations:** 1Centro de Investigación en Mecatrónica y Sistemas Interactivos—MIST, Universidad Indoamérica, Av. Machala y Sabanilla, Quito 170103, Ecuador; 2Universidad UTE, Av. Mariscal Sucre, Quito 170129, Ecuador; 3Department of Rehabilitation Medicine, Erasmus MC, 3000 CA Rotterdam, The Netherlands; d.s.lemusperez@tudelft.nl (D.L.); h.vallery@tudelft.nl (H.V.); h.l.d.horemans@erasmusmc.nl (H.H.); 4Faculty of Mechanical, Maritime and Materials Engineering, Delft University of Technology, Mekelweg 2, 2628 CD Delft, The Netherlands; 5Centre for Automation and Robotics (CAR UPM-CSIC), Universidad Politécnica de Madrid, 28012 Madrid, Spain; alberto.brunete@upm.es (A.B.); miguel.hernando@upm.es (M.H.)

**Keywords:** gait analysis, gait kinematics, 3D cameras, inertial sensors, neurological disorders

## Abstract

Three-dimensional (3D) cameras used for gait assessment obviate the need for bodily markers or sensors, making them particularly interesting for clinical applications. Due to their limited field of view, their application has predominantly focused on evaluating gait patterns within short walking distances. However, assessment of gait consistency requires testing over a longer walking distance. The aim of this study is to validate the accuracy for gait assessment of a previously developed method that determines walking spatiotemporal parameters and kinematics measured with a 3D camera mounted on a mobile robot base (ROBOGait). Walking parameters measured with this system were compared with measurements with Xsens IMUs. The experiments were performed on a non-linear corridor of approximately 50 m, resembling the environment of a conventional rehabilitation facility. Eleven individuals exhibiting normal motor function were recruited to walk and to simulate gait patterns representative of common neurological conditions: Cerebral Palsy, Multiple Sclerosis, and Cerebellar Ataxia. Generalized estimating equations were used to determine statistical differences between the measurement systems and between walking conditions. When comparing walking parameters between paired measures of the systems, significant differences were found for eight out of 18 descriptors: range of motion (ROM) of trunk and pelvis tilt, maximum knee flexion in loading response, knee position at toe-off, stride length, step time, cadence; and stance duration. When analyzing how ROBOGait can distinguish simulated pathological gait from physiological gait, a mean accuracy of 70.4%, a sensitivity of 49.3%, and a specificity of 74.4% were found when compared with the Xsens system. The most important gait abnormalities related to the clinical conditions were successfully detected by ROBOGait. The descriptors that best distinguished simulated pathological walking from normal walking in both systems were step width and stride length. This study underscores the promising potential of 3D cameras and encourages exploring their use in clinical gait analysis.

## 1. Introduction

Many people with central neurological disorders experience difficulties walking, primarily due to impaired control of the muscles involved in walking. Loss of strength, spasticity, muscle shortening, contractures, and pain can also contribute to abnormal gait patterns. These gait abnormalities often necessitate targeted interventions to improve gait function and enhance an individual’s overall quality of life. Gait analysis can provide valuable information on various characteristics of the gait pattern. By analyzing and scoring these features, healthcare professionals can make targeted treatment choices and evaluate the effect of their treatment. Complicating this process, gait analysis in a clinical setting traditionally requires a dedicated gait lab using fixed motion capture systems. Recently, with the advancement of optical and image processing technology, 3D cameras have also been proposed for use in clinical gait analysis [1]. 3D cameras require neither markers nor sensors to be attached to the body, minimizing setup time and reducing discomfort for the individual undergoing assessment. 3D cameras proved to be accurate in gait analysis, mainly to capture kinematics in the sagittal plane [2]. However, due to their small range of view, they have only been used for treadmill gait analysis [3] or for the analysis of a few gait cycles during overground walking [4]. Assessment of gait consistency requires testing over a longer walking distance in order to measure gait variability, which is commonly associated with fall risk [5].

For longer walking distances, multi-camera configurations have been used. For example, in the study presented by Geerse et al. [6], the ability to adapt walking to environmental circumstances was tested using an interactive walkway composed of multiple Kinect cameras. However, such a stationary configuration still requires a dedicated space and the certainty of fixed camera positions. To resolve these shortcomings, our robotic system, ROBOGait [7], which features a 3D camera on a mobile base, is set to move along with an individual while capturing. With such a setup, walking can be analyzed in a non-lab environment, e.g., a hospital corridor, for a large number of steps. ROBOGait is a “follower” robot that implements a control strategy to maintain a desired distance from the person. This setup differs from “walker robots” [8], where physical human-robot interaction is necessary.

In a previous study, the accuracy of ROBOGait for the assessment of normal gait along a straight line was validated [9]. The aim of the present study was to assess ROBOGait’s performance for the evaluation of common pathological gait disorders on longer, non-straight trajectories, which are typical for clinical settings. For this purpose, three gait patterns representative of common neurological conditions were simulated: Spastic diplegic Cerebral Palsy (CP), Cerebellar Ataxia, and Multiple Sclerosis (MS). These disorders were chosen because there is clear evidence of how they affect the gait pattern and because the treatment of these gait disorders is often based on the results of a gait analysis. For example, typical gait disorders in spastic CP are toe-walking and increased knee flexion [10]. Cerebellar Ataxia is characterized by irregular wide-based gait due to impaired balance control [11], and MS is characterized by stiff knee gait and foot drop during swing [12].

To validate ROBOGait’s ability to assess gait patterns in physiological and simulated neurological conditions, we compared a number of gait characteristics with the commercially available and validated Xsens system, which relies on wireless inertial sensors. Specifically, we (1) directly compared joint kinematics and spatiotemporal descriptors between ROBOGait and Xsens, and (2) quantified the accuracy, sensitivity, and specificity of the ROBOGait system in distinguishing simulated pathological walking from normal walking, using Xsens as a reference.

## 2. Methods

### 2.1. Participants

This exploratory study was approved by the Human Research Ethics Committee of Delft University of Technology. The experiments were conducted in corridors similar to those in a clinic or hospital. Eleven volunteers participated, four male and seven female, with an average age of 28 ± 8 years, height of 173 ± 9.4 cm, and a mass of 71 ± 11.2 kg. During the experiments, participants wore a T-shirt of the appropriate size provided by Xsens. Each participant was provided with a set of mounting straps to attach the Xsens inertial sensors to the body. As part of the protocol, a member of the staff assisted each participant in placing the mounting straps on their bodies to securely attach the Xsens inertial sensors. As part of the protocol followed by Xsens to create an anthropometric model of the participant, body height and foot length were measured.

The experiments lasted, on average, two hours per participant. None of the participants had a gait disorder, but they all simulated the different gait impairments based on the clinical experience of the research team. In our study, impaired gait simulation was assumed to be scientifically admissible because only kinematic gait imitation is expected [13].

### 2.2. Instrumentation

The skeleton tracking task in the ROBOGait system was performed with an Orbbec Astra 3D camera (range 0.6 m–8.0 m, Field of View 60° horizontal × 49.5° vertical × 73° diagonal) using the 0.34.1 release of the Nuitrack SDK. For each trial, 15 joint trajectories were recorded with the robot-mounted 3D camera at a sampling rate of 30 Hz. The raw data was post-processed using neural networks trained by supervised learning from a certified Vicon system, as explained in [14].

A wireless Xsens Awinda system was used for comparison (Xsens Technologies B.V., Enschede, The Netherlands). Fifteen inertial sensors were attached with velcro at strategic locations on the body to measure orientation and motion of each body segment, according to the guidelines for a “full body no hands” setup [15]. Table 1 describes the positioning of the sensors.

Xsens MVN Analyze Pro 2021.0 software was used for data capture at a sampling rate of 60 Hz. The Xsens system was calibrated before each participant session following the N-pose procedure within the MVN Analyze Pro 2021.0 software. Only calibration results with a “good” quality indication were accepted.

To synchronize the Xsens system with ROBOGait, the Sync In port of the MVN Awinda Station was configured to receive a rising edge of 3.3 V from the Sync Out port of the robot using a Bayonet Neill-Concelman (BNC) adapter. Once the participant started walking forward, the robot immediately followed, maintaining a preset 2.5 m distance in front of the participant. From then on, the participant followed the trajectory of the robot. Figure 1a shows a participant walking together with the robot in the experimental environment.

### 2.3. Conditions and Trajectory

Each participant walked in four different conditions: normal walking, simulated spastic diplegic Cerebral Palsy (CP), Ataxia, and Multiple Sclerosis (MS) walking. Spastic diplegic CP was characterized by toe walking, increased hip and knee flexion, and hip endorotation. Cerebellar ataxia in walking was characterized by poor movement coordination and balance. MS walking was characterized by a paretic gait with a drop foot. All simulated pathological walking was intended to be symmetrical. The participants were trained by an expert working in a clinical gait lab to simulate these pathological gait patterns and performed practice trials in advance. For each condition, one trial was performed by each participant.

To generate the walking trajectory, a map (20 m wide × 30 m long) of the environment was constructed, and a trajectory of about 50 m in length was designed. The mapping functions and navigation strategies to implement the walking experiments with the ROBOGait system are described in [7]. Figure 1b shows the planned trajectory and an example of the actual trajectories of the robot and a participant during one of the experiments.

### 2.4. Data Processing

The validation of the inverse kinematic process for calculating joint angles and the gait model used by the ROBOGait system have been presented in a previous study [9]. Joint kinematics from the Xsens system were directly calculated by the Xsens MVN Analyze software [16] using definitions based on ISB recommendations [17].

In this study, gait analysis of one side of the body was prioritized for a comprehensive assessment. Eighteen descriptors considered important for distinguishing pathological from normal gait were selected for the analysis. This set of descriptors was derived from recommendations by Molina and Carratalá [18], and consisted of twelve kinematic and six spatiotemporal descriptors. From the kinematic descriptors, six presented joint range of motion (ROM) during a gait cycle, and six presented joint angles during specific gait cycle phases. For both systems, strides were split using heel strike and toe-off events. These events were approached using the maximal anterior-posterior distance between ankle joints, as recommended by Zeni et al. [19].

ROM for trunk tilt, pelvis tilt and rotation, hip flexion/extension and abduction/adduction were calculated as the difference between the maximal and minimal angle during a stride. Knee flexion/extension ROM was calculated as the difference between knee maximal flexion during swing and knee position at initial contact. Temporal descriptors were calculated for both systems from the determined heel strike and toe-off events. The sagittal *x*-component of the relative position vector between both ankles was used as an estimation of stride length, while the frontal *y*-component was used as an estimation of step width. Since the world coordinate system for Xsens is static, the ankle coordinates were previously rotated according to the rotation of the pelvis around the vertical axis *z*, to align the *x* and *y* axes with the sagittal and frontal anatomical axes, respectively. This allowed us to align the vector between the ankles with the walking direction.

### 2.5. Data Analysis and Statistics

The motion capture of the robot-mounted 3D camera was sometimes affected by sunlight entering through the windows of the building, resulting in information about particular strides being lost. When this occurred, the information about the respective strides was discarded for both systems. It was measured that a total of 7% of all strides (gait cycles) collected during the experiments were lost for this reason. This, together with the variability of walking speed between conditions, led to a variation in the total number of strides analyzed per condition. The total number of strides used for the analysis was 2046, distributed as follows: 462 (22.6%) for normal, 554 (27.1%) for CP, 498 (24.3%) for Ataxia, and 532 (26.0%) for MS.

To evaluate direct differences between the two systems across all conditions and to flag significant differences between simulated pathological and normal walking within systems, generalized estimating equations (GEE) analyses for repeated measurements with exchangeable correlation structures were performed (IBM SPSS Statistics 26). GEE analysis is a semiparametric method that does not largely depend on the specification of the underlying distribution of the outcomes. As part of each model, a covariance matrix is estimated that represents the within-subject dependencies in repeated measurements, which also takes into account that some data could be missing. Assuming that data are missing at random, data imputation is unnecessary in GEE analysis. In contrast to complete case analyses, all observations are taken into account. As such, the GEE model is able to handle the unequal number of strides per participant and condition and uses all available data. System (ROBOGait and Xsens), condition (normal, CP, Ataxia, and MS gait), and their interaction were defined as predictors. Stride number was defined as a within-participant factor. The 18 gait descriptors were defined as separate dependent variables. In post-hoc analyses, the differences between the systems for each condition will be analyzed in pairwise comparisons using Wald tests with Bonferroni correction. Correcting for multiple comparisons is a common practice in statistical analysis to reduce the likelihood of obtaining false-positive results when conducting multiple hypothesis tests. One commonly used method is the Bonferroni correction, which involves dividing the desired significance level by the number of comparisons being made. Following this method, the significance level was set to p<0.003 (coming from 0.05/18) to correct for multiple testing. Readers can access the data and statistical analysis through the Appendix A.

To assess the performance of ROBOGait to distinguish pathological from normal gait, significant differences detected by Xsens between normal and simulated pathological gait represented a positive class (P), while the lack of difference represented a negative class (N). In this way, it was possible to determine the True Positive (TP), False Positive (FP), False Negative (FN), and True Negative (TN) parameters for the evaluation of specificity, sensitivity, and accuracy of the ROBOGait system when compared with the Xsens system.

Accuracy takes into account both changed and unchanged descriptors when comparing normal and simulated conditions and gives a general idea of the ROBOGait’s performance. Sensitivity, or true positive rate, represents the rate at which ROBOGait detects significant differences in descriptors in the same way that the Xsens system does. Specificity, or true negative rate, represents the rate at which ROBOGait determines non-significant differences in the same way as the Xsens system. The applied formulas are detailed in Table 2. Performance was considered better as accuracy, sensitivity, and specificity levels were closer to 100%.

## 3. Results

An example of knee joint kinematics measured by XSens and RobotGait, respectively, is plotted in Figure 2, and results of the GEE analyses are shown in Figure 3.

A comparison of walking parameters between paired measures of the systems, without distinguishing by gait condition, revealed that ROBOGait equally monitored 10 out of 18 descriptors, including pelvis rotation range of motion (ROM), hip and knee ROM, hip maximum extension in stance, hip maximum flexion in swing, knee flexion at initial contact, knee maximum flexion in swing, step width, and stride time. Relatively more differences were observed between the systems for spatiotemporal parameters compared with kinematic parameters.

Significant differences (p<0.003) were found for the remaining eight descriptors: ROM of trunk and pelvis tilt showed mean differences of −2 and −1 degrees, respectively; maximum knee flexion in loading response showed a difference of 5 degrees; knee position at toe-off presented an 8-degree difference; stride length differed by 0.12 m; step time showed a difference of −0.01 s; cadence differed by 1.7 steps/min, and stance duration had a difference of 1%, with a negative sign indicating lower values for ROBOGait.

Regarding how ROBOGait distinguishes simulated pathological from normal gait, Table 2 presents the number of times that both systems equally reported the presence or absence of significant differences between walking conditions. The results demonstrated that the accuracy of the ROBOGait system was good for MS, with 15 out of 18 descriptors correctly identified. For CP and Ataxia, 13 out of 18 and 10 out of 18 descriptors were correctly identified, respectively. This resulted in an acceptable average accuracy of 70.4% for the ROBOGait system. Additionally, sensitivity was found to be good for CP (True Positive (TP) = 10, False Negative (FN) = 3) and poor for MS and Ataxia (TP = 1, FN = 2; TP = 3, FN = 5, respectively). Specificity was good for MS (True Negative (TN) = 14, False Positive (FP) = 1 descriptors) and acceptable for CP and Ataxia (TN = 3, FP = 2; TN = 7, FP = 3, respectively). Based on these findings, the ROBOGait system exhibited an average sensitivity of 49.3% and an average specificity of 74.4%. The descriptors that best distinguished simulated pathological walking from normal walking in both systems were step width and stride length.

## 4. Discussion

### 4.1. Interpretation of Results

In this study, based on GEE analysis, eight out of the 18 descriptors analyzed showed significant differences during the direct comparison between ROBOGait and Xsens. The differences in most descriptors are too small to be considered clinically relevant, but for maximum knee flexion during loading response, knee position at toe-off, and stride length, the differences are sufficiently large to have clinical implications. Analyzing maximum knee flexion during loading response, knee position at toe-off, and stride length is highly important when assessing gait in patients with CP, MS, and ataxia. For example, stride length has been found to be 73% shorter in children with CP compared with children with normal development [20]. Significantly decreased stride length has also been reported in CP [21] and Ataxia [22]. Evaluation of knee flexion in CP patients is important because increased muscle tone (spasticity), mainly affecting the hamstrings and calve muscles, can lead to increased knee flexion and impaired push-off during walking and other activities [23].

For MS patients, evaluation of knee kinematics is important for analyzing stiff knee gait and foot clearance issues, which significantly impact walking ability. In Ataxia, abnormal knee kinematics and increased stride length variability reflect impaired balance control and the extent of motor impairment. Understanding these gait characteristics is crucial for tailoring interventions, tracking progress, and improving overall functional mobility and quality of life in individuals with CP, MS, and Ataxia. Currently, the lack of precision to correctly detect these parameters remains a limitation of the Robogait system.

The found differences could be caused by inaccuracies in the ROBOGait motion capture system. The skeleton model of the 3D camera assumes only two markers per body segment, while conventional gait models [24] use at least three markers per segment. Differences in temporal descriptors and kinematics related to gait cycle phases, such as step time and knee position at toe-off, may also be caused by inaccuracies in gait event detection. ROBOGait, positions the ankle’s virtual landmark anterior to the malleoli, at the intersection of the leg and foot segments. A difference of 2.31 cm has been reported for the location of this ankle landmark compared with a Vicon system [25]. This explains the inaccuracy in the detection of the ankle joint in 3D cameras and may also explain the small but significant differences found for step time and stance duration with respect to the Xsens reference. Finally, the differences could also partially be attributed to inaccuracies in the Xsens system. Although Xsens was used as a reference in this study, it should be noted that its absolute angles may also differ from commonly used golden standards due to calibration offsets [26].

Regarding the performance of ROBOGait for detecting differences between normal and simulated pathological gait, large variability was found between and within the simulated conditions. A good-to-acceptable performance (average performance 70%) was found for discriminating CP gait. Some significant differences found by Xsens, such as knee ROM and step time, were shown as trend differences by ROBOGait. In contrast, poor-to-good performance was found for discriminating simulated MS. Although accuracy and specificity were high, sensitivity was poor. However, relatively few deviations in walking were detected by both systems. Therefore, for MS, other descriptors than those currently chosen for the study could be investigated, which may be more sensitive to picking up on these deviations. The performance for discriminating Ataxia was poor to moderate. Discrepancies between both systems were mainly found for kinematic descriptors, both with respect to ROM and related to gait phases. These discrepancies in detected deviations are probably related to the aforementioned differences between the systems.

For both systems, most changes in descriptors during simulated pathological gait correspond with the clinical pictures of the simulated pathologies. For instance, the main characteristics of CP gait [20], such as increased knee and hip flexion, lower stride length, and increased step width, were successfully detected by both systems. For MS gait, increased step width was detected by both systems. Increased step width is a typical consequence of reduced walking balance in MS [27]. Both systems also detected differences in descriptors typically related to Ataxia gait, such as increased step width and reduced stride length [22]. Disagreement was only found for stance duration (no difference for Xsens, shortened for ROBOGait). Some characteristic deviations related to Ataxia gait (longer stride and step time and lower cadence [22]) were neither detected by Xsens nor ROBOGait. Possibly, both systems have difficulties with this particular detection, but it could also be that Ataxia gait is difficult to simulate well in healthy participants.

### 4.2. Limitations and Recommendations

A major limitation of the study lies in the fact that it analyzed simulated pathological gait from healthy participants. An analysis based on the gait of individuals with real motor disorders is proposed as future work. Another limitation resides in the gait model of ROBOGait’s 3D camera, which uses only two landmarks per body segment to estimate the inverse kinematic of the joints. Although it is possible to estimate the inverse kinematics of joints using a limited number of landmarks, relying on only two landmarks per body segment can lead to certain limitations and possible errors in the analysis, such as overlooking segment deformations, difficulty in capturing dynamic movements, and ambiguity of joint angles.

It is also worth mentioning other limitations in the methodology followed in this study, in which kinematic analysis of the ankle has been excluded. The ankle joint has been omitted, as the camera data has proven to be inaccurate in assessing the toe. Similar limitations have been reported in other studies, such as refs. [28] or [2]. For this reason, descriptors were not included to detect toe walking and foot drop, which are typical gait deviations observed in CP and MS, respectively. In hardware aspects, due to the principle of operation of the 3D cameras that emit light to obtain depth measurements, the experiments must be performed while avoiding sunlight entering through the windows. Stereoscopic 3D cameras could be considered as a way to reduce these effects. This is an approach that is currently being tested in new versions of the ROBOGait platform.

Finally, it is important to note that the results presented in this study were analyzed at the group level and should not be extrapolated to individual assessments without consideration. The complexities and unique variables of each participant may considerably influence the applicability and validity of these conclusions in an individual context.

## 5. Conclusions

In contrast to gait analysis systems commonly used in laboratories, the ROBOGait system can collect human gait data from longer trajectories in a more natural and user-friendly way. When directly compared with an inertial sensor-based reference system, the performance of the ROBOGait system to extract kinematic and spatiotemporal descriptors was satisfactory. Ten out of 18 descriptors were successfully monitored. When comparing the ability to distinguish simulated pathological gait from normal gait, variability in performance was found depending on the simulated pathological condition. In general, acceptable values were found for accuracy and specificity. The most important characteristics of simulated pathological gait were sufficiently registered by ROBOGait. This study reveals the promising potential of robot-mounted 3D cameras and encourages the research community to continue exploring their use in clinical gait analysis.

## Figures and Tables

**Figure 1 sensors-23-06944-f001:**
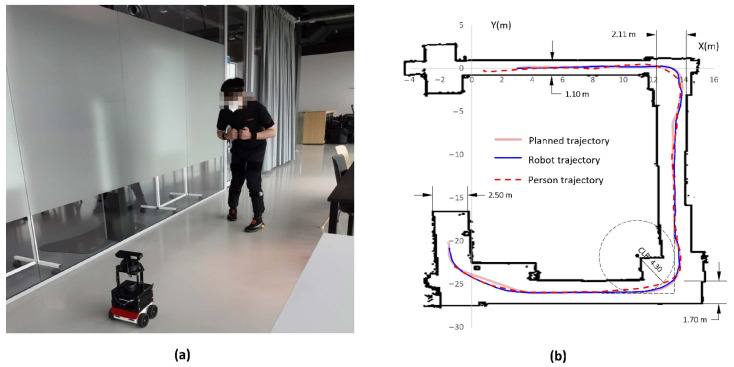
Overview of the study. (**a**) Experimental environment. (**b**) Experimental trajectory: planned trajectory (light red), actual trajectory followed by the robot (blue) and the trajectory followed by the participant (red dots). Corridor dimensions are shown along with the centerline radius (CLR) of the corners.

**Figure 2 sensors-23-06944-f002:**
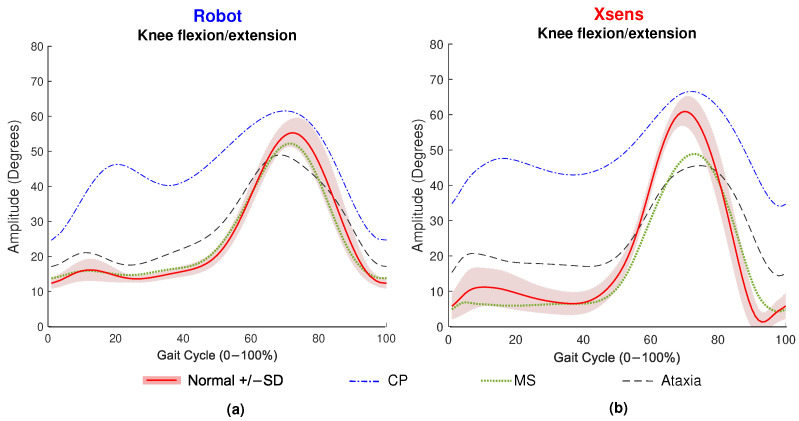
Knee kinematics as an example of gait patterns retrieved with both systems. Signals obtained with the ROBOGait system are shown in subfigure (**a**), and signals obtained with the Xsens system are shown in subfigure (**b**). The analysis includes Normal, CP, Ataxia, and MS conditions. Signals were averaged for all iterations performed for each condition.

**Figure 3 sensors-23-06944-f003:**
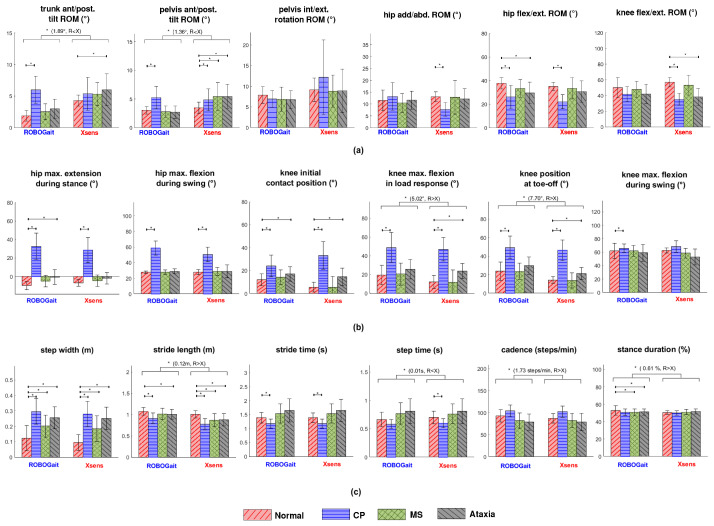
Kinematic and spatiotemporal descriptors of gait. (**a**) range of motion (ROM) descriptors, (**b**) non-ROM descriptors corresponding to instants of maxima and minima at important gait events. (**c**) spatiotemporal descriptors. The height of each bar represents the mean value, the whisker represents the standard deviation. Statistically significant differences based on GEE analysis between conditions (Normal, CP, MS, Ataxia) and between the ROBOGait (R) and Xsens (X) systems are shown. The significance level (*) was set to p<0.003 to correct for multiple testing.

**Table 1 sensors-23-06944-t001:** Positioning of the inertial sensors in “Full body no hands” suit configuration of MVN.

Body Segment	N. Trackers	Abbreviation	Position
Foot	2	FOOT	Middle of bridge of foot
Lower leg	2	LLEG	Flat on the shin bone
Upper leg	2	ULEG	Lateral side above knee
Pelvis	1	PELV	Flat on sacrum
Sternum	1	STERN	Flat, in the middle of the chest
Shoulder	2	SHOULD	Scapula (shoulder blades)
Upper arm	2	UARM	Lateral side above elbow
Fore arm	2	FARM	Lateral and flat side of the wrist
Head	1	HEAD	At the back of the head
Total of sensors	15		

**Table 2 sensors-23-06944-t002:** Performance of ROBOGait system when compared with the Xsens reference. TP: true positives; TN: true negatives; P: positives; N: negatives; FN: false negatives.

Measure	Value (%)				Formula
	CP	MS	Ataxia	Average	
Accuracy	72.2	83.3	55.6	70.4	(TP + TN)/(P + N)
Sensitivity	76.9	33.3	37.5	49.3	TP/(TP + FN)
Specificity	60.0	93.3	70.0	74.4	TN/(FP + TN)

## Data Availability

The data presented in this study are available in Appendix A.

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
