# Peer review of "Performance of a Mobile 3D Camera to Evaluate Simulated Pathological Gait in Practical Scenarios"

_sensors, 2023, doi:10.3390/s23156944_

Round 1

Reviewer 1 Report

sensors-2469008
1.    This manuscript presented gait analysis for normal individuals and simulated CP, MS and ataxia by RoboGait and Xsens. This manuscript offers substantial insights relevant to the field of gait analysis. However, to fully meet the journal's standards and to enhance clarity and comprehension for readers, some revisions are required before it can be accepted for publication.

Please see my comments for revising and improving the manuscript. Incorporating these suggestions will likely increase the manuscript's coherence, readability, and overall impact, aligning it more closely with the standards of Sensors.

Introduction
2.    Line 37-52. This paragraph is notably longer compared to the others in your manuscript.
In your objectives, make sure to highlight how your research aims to address or add to the existing body of knowledge on gait characteristics or differences in individuals with CP, MS, and Ataxia. This is critical because the reader might not be familiar with these terms or understand their implications on gait. By providing a concise background on these conditions and their effect on gait, you can help readers appreciate the relevance of your study more effectively.

Method
3.    Insufficient information about walking environment.
“To generate the walking trajectory, a map (20 m wide x 30 m long) of the environment was constructed and a trajectory of about 50 m in length was designed.”
In Figure 1b, it would greatly facilitate reader comprehension if you could provide a more detailed depiction of the walkway. Specifically, plotting the walkway's width and including additional dimensions such as the exact measurements of the width and the corner spaces could significantly improve the clarity of the representation. This will provide the reader with a more accurate understanding of the space within which the study was conducted.

4.    There was no mention of any statistical tests in the methods section. Please include the names of the statistical tests that you used in your analysis in the methods section. Also include the post hoc tests you used following the implementation of your main statistical tests.

Results
5.    I encountered difficulty in fully interpreting and reviewing this section due to the absence of named statistical tests. This omission hinders an understanding of the reliability and significance of your findings. Please rectify this in your revision.
6.    Figure 3, abbreviation R,X is not label.

Discussion.
7.    Lines 178-181: You have highlighted eight descriptors displayed differences between the two systems. While this information is relevant, it would significantly enhance the reader's understanding if you could delve deeper into the clinical implications of each individual descriptor. Please provide further interpretation and elucidation regarding how these descriptors are pertinent to the normal, CP, MS, and ataxia gait.

8.    For the sections spanning lines 33-36, 81-87, 191-193 and 226-236, it appears that your manuscript consists of several paragraphs that are particularly brief, often comprising only 1-2 sentences. While brevity can sometimes be beneficial, in this case, it hampers the flow and understanding of your manuscript. I would encourage you to revise these sections to create more comprehensive paragraphs.

9.    Throughout the manuscript, there seems to be an inconsistent use of the terms “subject” and “participant”. Please consider revising your manuscript to use only the term “participant”.

Please revise to avoid grammatical errors.

Reviewer 2 Report

This paper proposed “Performance of a mobile 3D camera to evaluate simulated pathological gait in practical scenarios”. The approach discussed in this manuscript is interesting. However, the following are some points that authors should be clarify or considered before the paper may be accepted.

1-     Carefully revise the abstract.

2-     In the literature review, I recommend incorporating additional recent and relevant references to strengthen your research background”

3-     In the experiment section, add more comparisons with your proposed method.

4-     Carefully recheck the abbreviations and define them before using; many referencs are missing in sections one and two.

5-     Even its not compulsory I suggest you upload your source code with revised manuscript.

6-     I strongly recommend proofreading your document, especially the literature section, to identify and correct any grammatical errors.

Extensive editing of English language required

Round 2

Reviewer 1 Report

sensors-2469008-peer-review-v2

Dear Authors, 

Thank you for your recent submission and revision of this manuscript. However, there are still some areas that require further attention and elaboration, particularly those related to the statistical analyses employed in this study.

By examining the results generated through SPSS (in Spanish output, which I could not thorough interpret), I infer that you have utilized a General Linear Regression model with stepwise selection. However, this has not been explicitly mentioned in the manuscript, and it is of utmost importance to clearly specify the type of test used. Please specify whether you employed multivariate analysis, ANOVA, or repeated measures ANOVA or others.

Furthermore, it is crucial to elucidate the post-hoc tests chosen following these main analyses. As post-hoc tests depend on the specific context and goals of your study, detailing your choice will provide better clarity for readers and reviewers.

To report the results of your statistical analyses, it is recommended to adhere to guidelines established by the American Psychological Association (APA) or the American Medical Association (AMA). These organizations provide a well-structured approach for presenting statistical data in an accessible and professional manner.

Given the complexities and nuances of statistical analyses, I suggest that you consult with a biostatistician to ensure the accurate execution and presentation of the study's statistical component. This step significantly enhances the reliability of the results and provides greater assurance of their interpretation.

Subsequently, please include the necessary information from the statistical analysis in results section accordingly.

Please consider these remarks and make the necessary adjustments.

NIL
